# Namaste care in the home setting: developing initial realist explanatory theories and uncovering unintended outcomes

Sonia Michelle Dalkin  ,[1,2] Monique Lhussier,[1,2] Nicola Kendall,[3] Joanne Atkinson,[1] Sharron Tolman[4]

[1]Faculty of Health and Life Sciences, Northumbria University, Newcastle upon Tyne, Tyne and Wear, UK
[2]Fuse (The Centre for Translational Research in Public Health), Newcastle Upon Tyne, UK
[3]St. Cuthbert's Hospice, Durham, UK
[4]Dementia UK, London, UK

**Correspondence to**
Dr Sonia Michelle Dalkin;
s.dalkin@northumbria.ac.uk

## ABSTRACT

**Introduction** The End-of-Life Namaste Care Program for People with Dementia, challenges the misconception that people with dementia are a 'shell'; it provides a holistic approach using the five senses, which can provide positive ways of communicating and emotional responses. It is proposed Namaste Care can improve communication and the relationships families and friends have with the person with dementia. Previously used in care homes, this study is the first to explore the pioneering use of Namaste Care in people's own homes.

**Objective** To develop initial programme theories detailing if, how and under which circumstances Namaste Care works when implemented at home.

**Design** A qualitative realist approach following the RAMESES II guidelines was employed to understand not only whether Namaste Care has positive outcomes, but also how these are generated, for whom they happen and in which circumstances.

**Setting** A hospice in the North East of England, operating in the community, through volunteers.

**Participants** Programme theories were developed from three focus groups with volunteers implementing Namaste Care (n=8; n=8; n=11) and eight interviews with family carers (n=8).

**Results** Four refined explanatory theories are presented: increasing engagement, respite for family carers, importance of matched volunteers and increasing social interaction. It was identified that while Namaste Care achieved some of the same goals in the home setting as it does in the care home setting, it could also function in a different way that promoted socialisation.

**Conclusions** Namaste Care provides holistic and personalised care to people with both moderate and advanced dementia, improving engagement and reducing social isolation. In the present study carers often chose to use Namaste Care sessions as respite. This was often linked to their frustration of the unavoidable dominance of task-focussed care in daily life. Individualised Namaste Care activities thus led to positive outcomes for both those with dementia and their carers.

## INTRODUCTION

Globally, the numbers of people living with dementia will increase from 50 million in

### Strengths and limitations of this study

► This study details the theory building process in realist evaluation.
► Theory *building* was focussed on as opposed to theory *testing*, due to the lack of current evidence surrounding the use of Namaste in the person's own home and the small participant numbers.
► The study uses focus groups and interviews to develop rigorous and transparent programme theories.
► A limitation of the study is the sample size; while some programme theories were not substantiated by the data, it could be that this was due to the limited sample size.

2018 to 152 million in 2050, a 204% increase.[1] Despite this, the WHO[2] recently highlighted that 146 countries currently do not have a national plan for Dementia. Those countries that do have policies often employ a holistic focus on care (eg,[3–5]) however as the disease progresses often the focus of care shifts toward the physical body.[6–9] This emphasis on physical needs often comes at the expense of personhood needs.[10]

Activity has been demonstrated to be a positive therapeutic intervention with potential to enhance quality of life and reduce behavioural symptoms in those with dementia, thus potentially avoiding pharmacological treatments.[11] There is an increasing body of research into non-pharmacological, psychosocial and community-based interventions and their impact on quality of life and well-being for people with dementia and their family members or carers.[12–14] Accordingly, the 2019 National Institute for Health and Care Excellence Guidance on Dementia refers to several activities that fit under the umbrella of psychosocial and non-interventions including aromatherapy, art, gardening, baking, reminiscence therapy, music therapy, mindfulness

and animal-assisted therapy.[15] Furthermore, the guidance suggests that the activities offered should be based on an understanding of that individual's unique set of life experiences, circumstances, preferences, strengths and needs.[15]

Meeting this brief is the 'The End-of-Life Namaste Care Program for People with Dementia (NC)'.[16] As dementia advances, family carers describe a changing relationship and sense of loss, which can cause significant distress.[17] Finding new ways of communicating is important to help the family carer and person with dementia to maintain a good quality of life. NC (http://www.namastecare.com/) challenges the perception that people with advanced dementia are a 'shell', a 'living death'; it provides a holistic approach based on the five senses. NC can improve communication and the relationships families and friends have with the person with dementia.[18] NC is a psychosocial intervention that has been implemented variably internationally;[18] research is beginning to develop understanding about the intervention and it's cost implications,[19–25] but to our knowledge has only been formally evaluated in care home settings. A hospice in the North East of England has made provisions to provide NC in the person's own home. This is operationalised through the training of volunteers who are then matched with a person with dementia, in terms of personality, abilities and interests, for example. Two specialist workers lead the project and orchestrate training, debrief events and matching of patients and volunteers. Volunteers visit the person for 20 sessions, which are usually weekly and last 2 hours. Delivery is therefore significantly different to that initially outlined by the NC originator, who suggests that it should be delivered two times per day, 7 days a week[26] (table 1). However, stakeholders in a recent review indicated that this was unlikely to be feasible in most care homes in the UK.[19] The review also found little empirical evidence on the optimal 'dose' of sensory interventions, such as NC, although the literature did suggest that interventions that are delivered more regularly are important for creating a sense of reassurance and familiarity and building trusting relationships between residents and carers. Home delivery of the intervention also differs significantly from care home delivery in terms of staff impact; use of NC in care homes is also intended to address staff satisfaction by enabling them to have quality time with residents that is not just focussed on task-based activities. However, there are similar implications for family members' in the delivery of NC in the home environment, as volunteers delivering NC encourage their participation. This would engage family members in quality time with their loved one, as opposed to task focussed care.

To our knowledge, the hospice included in this study is one of only two hospices in the UK implementing this type of model for NC; the other service is located in London.

Healthcare provision in Europe, the USA and Australia has seen an emphasis on providing people with choice around the location of their care and death, frequently with an emphasis on driving care into the community and facilitating home deaths.[27] Despite this, statistics indicate that home deaths in people with dementia are generally low internationally, with significant variance across countries reported as a product of variability in end-of-life care provision.[28] Furthermore, unmet needs are common in those with dementia living in the community, and most are non-medical.[29] Recent research has highlighted that home-based dementia care should identify and address unmet needs by focussing on both care recipients and caregivers to enable the person with dementia to remain at home.[29] With current policy driving care into the community, ways to support quality of life for people with dementia in their own homes is pivotal.

This research contributes in two ways to the NC nascent knowledge base. While research to date has demonstrated outcomes in care homes, little is yet understood about how and why they occur. While this study is focussed on delivery of NC in the person's own home, it will highlight

**Table 1** Summary of differences in delivery of NC in the residential care and home setting

| Residential care home | Person's own home |
| --- | --- |
| 7 days per week, 4 hours per day (2 hours in the morning, 2 hours in the afternoon) | 2 hour visits once a week |
| Varied care home staff carrying out the Namaste session | Consistent volunteer carrying out the Namaste session |
| Given the frequency of the session, this contributes considerably to the daily care of the resident, as well as hydration levels | Less frequent and so less direct contribution to care and hydration levels |
| Family most likely not present | Family present in the home and invited to learn about and participate in NC |
| Staff satisfaction targeted through improving relationships with residents through non-task focussed care | Family engagement targeted through invitation to participate in NC with volunteer and provide non-task focussed care |
| Option to have a dedicated space for NC (a Namaste Room or special area) | Requires creating a suitable environment/atmosphere within someone's home |
| Potentially unfamiliar surroundings | Familiar surroundings |

NC, The End-of-Life Namaste Care Program for People with Dementia.

pivotal contexts (not just related to physical location) and underlying mechanisms, which may also be relevant to the care home setting. The context and mechanisms identified in this research could warrant further research in the care home setting. Second, the unique implementation in a community setting affords the opportunity to explore the impact of the home as a novel intervention context.

## Objective
To develop initial programme theories detailing if, how and under which circumstances NC works when implemented at home.

## METHODS
Realist evaluation is a theory-driven approach which seeks to understand not only whether an intervention works, but what it is about it that works, for whom, in what circumstances and why.[30] It acknowledges that interventions take place within complex social systems[31] and is therefore well suited to studying interventions such as NC.

The formulae Context+Mechanism=Outcome (C+M=O) is used to express generative causation in realist reserach, with mechanisms consisting of both intervention resources and stakeholder reasoning.[32] An intervention offers resources (Mechanism resource: such as hand massage, for example) which can alter the context into which it is introduced[32] (C; the person with dementia is experiencing restlessness and agitation), triggering a change in the reasoning of intervention participants (Mechanism reasoning (M); patient relaxes and feels more able to engage), leading to a particular outcome (O; the person with dementia is less agitated potentially avoiding a respite admission). CMO configurations are used as explanatory formulae (otherwise referred to as realist programme theories), which are developed and refined with empirical data. As with other evaluations of person-centred interventions,[33] the use of a realist approach will help to expose the multiple resources delivered as part of NC, the ways that these may be employed with different people, in diverse situations and how these generate outcomes. Applying the principles of realist evaluation therefore will determine why NC is successful or unsuccessful, in particular contexts.

## Patient and public involvement
Due to the small-scale nature and limited funding of the research, patients and the public were not involved in the development of the research question or design of the study. Members of the public from the hospice were consulted on dissemination plans.

## Operationalisation of the study
A realist approach was operationalised in two phases following the RAMESES II[34] guidelines: phase 1 focussed on building programme theories with volunteers

implementing NC in the community, using focus groups (n=3, with 8, 8 and 11 participants, respectively, 1 male in each focus group); phase 2 consisted of refining the theories with family carers of people who had received the NC intervention (n=8, 6 male, 2 female). Focus groups took place at the hospice and interviews were conducted either at the hospice (n=1) or at the family home (n=7). All focus groups and interviews were digitally recorded. Participants were recruited through the Namaste Leads. Volunteers and family carers had the study explained to them in person by the Namaste Lead; if they were interested in participating they provided their email address and/or telephone number with permission for it to be given to the lead researcher (SMD). SMD then contacted potential participants to arrange a suitable time and location for interview (family carers) or provided the date and time of the focus group (volunteers). Participation in the focus groups and interviews was voluntary and attendance at one focus group did not assume attendance at subsequent ones. Participants were not given any remuneration for the participation in the study, although volunteers were provided with lunch at the focus groups.

## Setting and referrals
The hospice is set in the North East of England and covers two areas, one town (population of around 25 000) and one city (population of around 65 000). The hospice delivering Namaste in the community was founded in 1988 and is a registered charity, which also receives some income from the National Health Service.

Family carers self-referred to the hospice to request access to NC. They were then matched with a trained volunteer. The hospice received requests for NC from family carers of people with severe and milder dementia. In order to be inclusive, as a community intervention, the hospice provided NC to all, not just to those with advanced dementia. Referral criteria is provided as online supplementary information 1. All family carers currently engaged with NC at the hospice at the time of the study were invited to participate, by telephone call conducted by one of the NC Leads. Before interviews with family carers could be conducted, their loved one must have experienced a minimum of four NC sessions. This requirement, combined with the hospice's referral criteria constituted the inclusion and exclusion criteria for the study.

Volunteers all began training in NC 3 months prior to the study beginning. They were introduced to their matched person with dementia and their carer through the hospice NC Lead at the person's home. During this informal meeting, the Life Story of the person with dementia was discussed, in the form of a larger document called 'My Namaste Care'. This formed a starting point for creating personalised care based on sensory interactions. This was a key step in matching personalities, histories and interests, which was thought to be significant to the intervention success. Volunteers then visited the person with dementia 20 times, in their own home, once per week for 2 hours.

Should volunteers encounter issues they reported immediately to one of the two NC leads, one of whom was a trained Admiral Nurse. Issues raised with the NC leads included nursing related concerns (eg, pressure sores) or queries about NC delivery (eg, asking permission to use a different approach, such as going outside). NC sessions were personalised based on the person's 'Life Story', which was completed before NC sessions began by the NC Lead and shared with the volunteer. All sessions included multisensory bespoke activities such as hand massage, aromatherapy and music in those with more advanced dementia, and exploring the garden, baking and singing in those with milder dementia.

Data was transcribed verbatim and imported into NVivo. A realist logic of analysis employing CMOC (context, mechanism, outcome configuration) was used to build and refine programme theory.[35] Throughout the evaluation, analysis moved iteratively from particular examples, to refinement of programme theory, use of substantive (or middle range) theory and further iterative data collection. This continuous loop of analysis generates a reflexive process, utilising retroduction to spark insight and develop meaning. Retroduction uses both inductive and deductive logic, as well as insights or hunches to identify hidden causal forces that lie behind identified patterns or changes in those patterns.[36] The iterative approach adopted in realist evaluation allows the revisiting of the data as new additional questions emerge and connections are established, thus deepening the understanding and meaning of the findings.[37]

Volunteers are referred to throughout analysis as V1 to V12, and family carers as P1 to P8. The source of the data is indicated using Focus Group (FG) and then the number of the Focus Group (out of three). For example, the first focus group is referred to as 'FG1'.

### Findings
The findings are presented following the phases of the research.

### Phase 1: building programme theories
#### Impact on people with dementia
The Life Story was part of a larger document called 'My Namaste Care' and formed a starting point for creating personalised care based on sensory interactions.

**V1, FG1:** It's called My NC. So it's like a life-story template that we use. With, sort of, prompt questions that we work through. But it's capturing those really special memories that might ignite some kind of recognition.

The life story was thought to be a key intervention component, although volunteers suggested that it was only a basis to work from.

**V1, FG1:** There's the things that you plan from the life story. (…) sometimes you don't know what's going to work. So an example, I took some vintage rose body spray stuff to try this week, and I don't know

that she likes it. And this female is not speaking at all now, so I let her smell it. And clear as anything – 'Oh, nice…' Was the response I got.

It is also important to acknowledge that reactions of the person with dementia to NC stimuli are not always predictable.

**V2, FG3:** So, the female that I visit, she's been quite static, really, for the time I've been visiting. There's times I try things and I don't get much of a response, and then there's other times I get a really lovely response.

Therefore the life story created a base for volunteers to work from, leading to experimentation with different resources which could engage the person, in ways that had previously become difficult.

**V4, FG3:** Some days, she's needed very little prompting. I mean, we made 12 cupcakes, 1 week, and she iced them completely on her own.

Once the person was engaged in sensory activities as part of NC, often a response was observed by the volunteer.

**V2, FG3:** You might see a difference from her being fairly tense in how she is in her body to being more relaxed… Increased eye contact across the time, from the beginning of a session to the end. You do see changes like that. But they're quite difficult to measure, I think.

Those who had more advanced dementia also indicated engagement and an emotional response, even if verbal communication was not possible. The volunteers were skilled at picking up non-verbal responses to the intervention.

**V2, FG3:** You know, it's about getting to know the person. She tells me a lot, just with our non-verbals. I was reading this poem (…) There was lots and lots of, sort of, film star names that I was reading out as part of this poem. And when we got to Marlon Brando, she was like this… (wide excited eyes) And when I checked out with her husband, sure enough, he was her favourite. So, she was still telling me. She was still communicating in her own way.

From the findings presented above, the following programme theory was developed:

**Programme theory 1:** The volunteer is aware of the person's life story (context). Experimentation based on the life story is used to identify useful personalised activities (resource) which evoke an emotional response from the person with dementia, meaning they engage with the NC worker (reasoning). The outcome can be relaxation, engagement, increase in alertness or emotional response.

#### Impact on family carers
One of the guiding principles of NC is to engage the people surrounding the person with dementia, whether this be

care home staff or family carers. Volunteers suggested that often family carers felt that they had no hope and felt a sense of helplessness, which was compounded by a lack of support.

**V1, FG1:** You know, a lot of people talk now about where they go through the medical system, and there's a lot of… It's a very impersonal feeling a lot of the time. Not necessarily with general practitioners, but with going through the hospital system and… You know, it's just… Next. So whether it's just that very different, personal feel [with NC], it just seems to be very meaningful for people.

In this context, volunteers believed that the weekly visit by the NC volunteer had a significant impact on family carers too, offering acknowledgement, support and hope:

**V6, FG1:** Families don't like the idea that there's no hope anymore… They hate the phrase 'There's nothing that can be done.' They really don't like that. So I think for some families, that sense of hope that actually there is something that you can do [NC]. You are… Somehow, bringing something very positive to that person.

Such reactions from loved ones led to the realisation that the person is still living, thus challenging the idea that those with dementia are a 'living shell', which often led to a feeling of increased hope and well-being for family carers.

**V4, FG1:** So, maybe, seeing that patient smile reminds the husband that, you know, she's still in there. Or, you know, laughing or… Or whatever. I mean, just the, sort of, difference between the… There's a tendency to think the emotional piece has died with the cognitive.

**V2, FG3:** And I think he also just enjoys seeing her enjoying herself.

This inherently acknowledges that the person is capable of feeling, expressing and engaging, even if differently than before. Impact on family carers is thus mediated through this valuing of the person with dementia, and the close bond they have with them.

However, volunteers were wary of providing what could be thought of as too much hope, being conscious of the potential for family carers to misconstrue or overestimate the potential impact of NC.

**V2, FG3:** I think it does give them a little bit of hope. The husband of the female that I visit – that's been a bit of a problem (…) unrealistic expectations, initially. So he was asking if I was going to get her talking again and that kind of thing. So, I think you've got to tread carefully with that.

Following the analysis above, programme theory 2a was built:

**Programme theory 2a:** In a context where family carers have seen their loved one decline and been told there is 'no hope' and received little or impersonal care (context) use of NC to evoke reactions from their loved one (resource) leads to them feeling hopeful and acknowledging that their loved one is still 'living' (reasoning). This leads to increased hope (outcome) and well-being.

An additional context was highlighted at this point; family carers were often focussed on task-based daily care (around cleaning and feeding, for example) which took up a great proportion of their time and energy. This meant that they sometimes struggled to engage with NC, as initially expected.

**V1, FG1:** So it is down to one main carer, often, to do a lot of the… And it does become very functional, very task-based.

**V6, FG2:** The husband of the person I see, (…) he asked once how things went (…) And he said he felt a little guilty, like 'It's not something I have had time to do' or something like that. And I thought later that what I should have said is, you know, you do everything else. And this is icing on the cake or something. But he expressed this… It wasn't jealousy or anything, it was like just… You know, wishing that he had.

Family carers could enjoy respite because the person focussed quality of the NC approach meant that the family carer felt that their family member was in safe hands, offering a level of engagement that they themselves could not always achieve.

**V6, FG2:** He's… A couple of days have been sunny and beautiful. And he's very interested in his garden. So he loved the idea that, you know, she was being stimulated and cared for. And he could escape to the garden.

**V4, FG3:** Occasionally, if she's having a foot massage, he will sit on the sofa and contribute. But, most of the time, he'll take himself off to do the ironing or his crossword – just, sort of, upstairs. And he said that he benefits from that little 2 hour slot of respite.

As a result of this analysis, a rival programme theory was created:

**Programme theory 2b:** Family carers provide task-focussed care and have little input from other services (context). A familiar NC volunteer provides 2 hours of interaction with the person with dementia (resource) which eases off worries about the family carer's loved one and allows them to have some respite (reasoning) which leads to an increase in carer well-being (outcome).

### Family carer use of NC

As described above, volunteers described how they felt often family carers roles had become task focussed, as opposed to engaging in enjoyable activities with their loved one. This was despite volunteers offering participation in NC to family members. Family carers had shown

initial interest in NC, but at this point volunteers assumed that they had not been confident enough to use the techniques themselves.

**V2, FG3:** I've noticed her husband coming in more and more and more. You know, having… You know (…) and I'm showing him what I'm doing and he's showing more interest. I don't know whether he would ever be confident enough to try it himself.

The volunteers suggested that if the NC box, which contains all of the items the volunteer uses with the person with dementia (eg, music, hand creams) was left in their home, family carers may become familiar with it and potentially use some of the techniques introduced by the volunteer. This would enable them to engage with the person on a different level than purely task focussed.

**V7, FG1:** I think for some families it'll help take away the, sort of, pure task-focussed work. You know, that we have to do every day. The washing, dressing and the, sort of, general day care… Day-to-day care (…). I think some families… I can see that opening up to them, to a different view of… Of the way they care for the person.

The analysis above resulted in the following programme theory 3:

**Programme theory 3:** Being often task focussed, family carers recognise the value of NC (context). A tailored activity box is left at the person's home (resource). The family are keen to engage in activities that enable them to connect emotionally with their loved one (reasoning). Family use NC independently (outcome).

### 'One-on-one' use of NC with matched volunteer

Matched 'one-on-one' volunteer time, continuity and consequently relationship building were identified as a key feature of using NC at home, as opposed to offering it in a group environment at the hospice, as part of their adult day care provision. This was considered as a positive of the home environment as opposed to the traditional use of NC in a care home, where a group environment is employed.

**V1, FG2:** And I suppose you're getting the same person, as well. So you have got that ability to build the relationship

Matching volunteers with the person with dementia and allowing them to have one to one sessions regularly resulted in the volunteers understanding the person's likes and dislikes despite often limited verbal abilities.

**V1, FG1:** I mean, this was probably about week four or five of visits. So I sort of know, roughly, what… What relaxes her. So I know a hand massage, she'll get quite sleepy and relaxed. What I've learnt is that if I kind of joke around, that brightens her up. You know, you get a response that way. So it is based a little

bit on, sort of, observing across the weeks what she, sort of, engages with.

It was also evident that volunteers built up a very strong emotional connection with the person they were matched with.

**V4, FG3:** And she used my name for the first time, yeah. On Wednesday. Which was heart-warming (crying).

This strong emotional connection in some cases resulted in recognition of the volunteer by the person with dementia.

**V1, FG1:** It feels like there's some recognition there (…) she recognises how I… How she feels when I'm there. So that emotional connection is what… Is, sort of, the link between each week.

This can evoke reactions and a proactivity that might have been largely unseen before. Furthermore, recognition also transcended the place related context of the NC intervention (V10).

**V9, FG3:** Well, I wash my female's feet every week. She doesn't like her hands to be washed, but she loves to put her feet in water. And, at first, I would say, 'I'm just going to get the dish, you know…' But now I pick the dish up and when I come back her socks and shoes are off. She's taking them off.

**V10, FG3:** Well, my female is going to respite, because her husband has been taken into hospital. So, I went to visit her yesterday, and I didn't know if she would recognise me in a different situation – but she did, straightaway. And she kept saying, over and over, I'm so glad you came.

The following programme theory was built based on the analysis presented above:

**Programme theory 4:** One volunteer is aligned to a person with dementia and spends 2 hours per week solely with that person (context). The volunteer therefore has a knowledge history of what works/doesn't work and what the person likes (resources). This allows the volunteer and the person with dementia to develop a strong emotional connection (reasoning). The outcome is an increased engagement which might have previously been thought of as impossible (outcome).

The focus groups with NC volunteers led to the formulation of four programme theories, which focussed on: (1) the life story, (2) hope for family carers, (3) the development of new ways of interacting, (4) the relationship between the volunteer and the person with dementia. These initial programme theories were then refined through interviews with family carers.

### Phase 2: refining and testing programme theories
Phase 2 consisted of interviews with family carers of those with dementia who were engaged in NC sessions.

Programme theory 1, which focussed on the direct response of the person with dementia to the NC interventions, in the context of good knowledge of the person's life story, was well supported by the interviews with family carers.

**P4:** Because they've done their Life Story. You see… My dad (…) liked his music with church. So, (Volunteer) has come along with… From the sport point of view. Music from Grandstand and, you know… Some of those. But also he's found You'll Never Walk Alone, which is… Although it is music, it's what they used to sing at the church. And just played it off his tablet. They were all singing. My mam and (Volunteer) were singing to him.

However, family carers also indicated the importance of social interaction between the person with dementia and the NC volunteer. This was particularly important, but not limited to, those with less advanced dementia.

**P3:** I think it's valuable. I think it's worthwhile. And I think (Person with dementia's Name) definitely gets something out of it, because I think she desperately needs that interaction with people.

**P4:** Well, I mean, in the home, like my mam – who will not go out – you're taking away an element of isolation. You're bringing an interest from outside into her. Which she wouldn't get.

While volunteers emphasised the need to trigger an emotional connection with the person receiving NC, regardless of their verbal abilities, family carers talked more about the value of social interaction. One family carer in particular questioned whether it was specifically interaction with the NC volunteer that was important, or whether it was just social interaction in general.

**P8:** I think she just enjoys any interaction, to be quite honest.

**Refined programme theory 1:** As dementia progresses, people's opportunities to engage in social interactions that are meaningful to them become more limited (context). Using their knowledge of the person's life story to develop a set of bespoke interactional tools and techniques (resources), NC volunteers evoke an emotional response in the person with dementia (reasoning), leading to a set of relaxation, engagement and alertness outcomes.

Programme theory 2a confirmed that family carers often felt a lack of hope and helplessness about their loved ones dementia, but the theory was less well supported in terms of NC increasing that hope through interaction. Family carers indicated that they still found it very difficult to interact with their loved one, and struggled not to see them through the same lens as they did when they were well.

**P3:** I can't react to (my wife) the way that a stranger does anymore. I do my best to react, and interact, with her — to look after her and all the rest of it. But I'm her carer. I find it… It's not easy for me to, sort of, like keep on talking to (my wife).

Some family carers went so far as to think that it was not possible for anyone to communicate with their loved one, as they believed dementia prevented this.

**P7:** So, there's no communication. I can't communicate with him. I couldn't ask him… You can ask him if he has a… He scrunches his face, or if he cries out, if you ask him what's wrong, have you got a pain, he doesn't know. He doesn't know whether he's got a pain. So, therefore there's nobody can communicate with him.

**Programme theory 2a** was formulated as: In a context where family carers have seen their loved one decline and been told there is 'no hope'/'nothing can be done' and received 'impersonal care' (context) use of NC to promote reactions from their loved one (resource) leads to them feeling hopeful and acknowledging that their loved one is still 'living' (reasoning). This leads to increased hope (outcome) and well-being.

Consistent with realist analysis, where theories are refined, substantiated or rejected as they are tested through empirical data, the lack of substantiation of this theory led to its rejection at this stage. Support was found for the alternative programme theory 2b though, which related to the use of NC as respite for family carers.

**P2:** It's continuous, basically, when you're looking after somebody with Alzheimer's. You know, it's 24… Well, not quite 24/7, but a lot of the time. And it's just nice to have a couple of hours to do something completely different, you know. And know that they're in safe hands.

One family carer also felt that her not being present was an advantage as it engaged her mother more in the NC sessions.

**P4:** I think it's nice for mam, me not being involved. Because, if I'm there, mam will look at me to answer questions. Will look at me to make conversation. So, I'm better out of the way. It means she has to… And she starts talking. So, yeah, I potter on.

The 2 hour respite provided by NC sessions was particularly appreciated in light of the perceived lack of services to help people with dementia and their family carers.

**P4:** But I am literally on duty until I get dad into bed, and his last eye drops in – that's usually about quarter to 10 at night. And that's 7 days a week (…) Because I can't take holidays, I can't have breaks. I get 2 hours (official respite), once a fortnight.

**P7:** I've had no help whatsoever (…) They say on there (TV), there's people to get help. They don't… You're just left. I mean, I was just left to manage on my own…

As a result of the analysis, **refined programme theory 2b** was postulated: Family carers provide continuous care and have little input from other services (context), provision of 2 hours contact with a trained NC volunteer (resource) allows them to concentrate on other things, knowing that their family member is in safe hands (reasoning) which gives them restorative time and space (outcome).

No support was found for programme theory 3, which suggested that NC would engage family carers and give them knowledge of how to engage in sensory activities with the person with dementia.

**P6:** And, of course, I want to think they've played music and read poetry and massaged my wife with cream on their hands… Because what my problem has been – I can take care of her physically… I can keep her safe, I can keep her warm, I can keep her dressed and comfortable… But I can do nothing at all to improve the quality of her life, you see.

One family carer also suggested that she thought her mother would feel uncomfortable if she were to try to use the techniques herself, as she already provided so much care for her, which was time and resource intensive.

**P4:** They have the time to spend to really draw them out. I haven't. I've got to break off to go and do their meals, to get the washing dried… So, it's nice that somebody has the time to spend with them, and solely them. And mam and dad accept that. When they're not there, they wouldn't do that with me.

**Programme theory 3** was therefore not supported by the family carers' interviews.

**Programme theory 4** concerned the importance of having one volunteer aligned to one person with dementia for the 20 sessions of NC.

**P2:** She got quite emotional herself. You know, which was nice. I mean… She obviously cared that much, you know. And, yes, we did very much see her as a friend.

Family carers echoed the focus group discussions describing a very strong emotional connection between the person with dementia and the NC volunteer. This was often described using recognition as a proxy.

**P3:** And I think it's just, you know, spending time with her. Because her eyes do light up, mind, when (Volunteer) comes. So, there is some sort of recognition. As almost, like, a friend or relative so… I think from that point of view, that makes me happy.

Related to the importance of the emotional connection, family carers highlighted the importance of having a consistent NC volunteer.

**P4:** If you just… One person stops and another person comes in, I think you're then going to have a

knock-on effect that it's going to take, again, two, three sessions before you have the relaxed atmosphere again.

**Programme theory 4** was therefore supported, stating that: One volunteer is aligned to a person with dementia and spends 2 hours per week solely with that person (context). The volunteer therefore has a knowledge history of what works/what doesn't work and what the person likes (resources). This allows the volunteer and the person with dementia to develop a strong emotional connection (reasoning). The outcome could be considered as the recognition of the volunteer by the person, but actually this leads to friendship, which could suggest an increased quality of life for both people.

Interviews with family carers highlighted the importance of the one-to-one interaction in NC. In a care home setting, NC is usually implemented in a group environment. Family carers discussed group environments in relation to other activities they had tried with their loved ones, or group family situations:

**P4:** Although I tried to persuade her to go to, like, the dementia cafes or singing for the brain and all this type of… No. Won't go.

Discussions were also then framed to ask about NC in a group environment, as is delivered in care homes:

**P2:** Yeah, it's far more focussed [in own home]. It's focussed on the individual. Plus the fact that in general, quite willingly, she's passive in a big group. She has the rest of the group, you know, to take over basically. And so she doesn't contribute. Not that she, sort of, doesn't want to. She just doesn't feel the need to, if you see what I mean? She doesn't feel, sort of, overawed by the group.

Family carers also described how they liked their loved one to have social interaction, as described in programme theory 1, but often it caused the person anguish.

**P8:** I think the thing with (my wife) is it's got to be one-on-one. That really… It's sort of the experience with her – if there was time to leave her in a group situation… It would just upset her so much. And I think… I think she thinks to herself, why am I here with these people, who I don't know, and there's something wrong with them.

The interviews with family carers led to refinement of theory 1 (the life story), rejection of theory 2a (hope for family carers) and further development of theory 2b (respite). Theory 3 (development of new ways of interacting) was also rejected, but support was identified for theory 4 (relationship between the volunteer and the person with dementia). The one-on-one delivery of NC in the home setting was also highlighted by family carers.

**Table 2** Programme theories developed and their refined counterparts

| Programme theory | Focus group developed theories | Interview refined programme theories |
|---|---|---|
| | Refinements indicated in bold italics | |
| 1. | ***The volunteer is aware of the person's life story (context). Experimentation based on the life story is used to identify useful personalised activities (resource)*** which evoke an emotional response from the person with dementia, meaning they engage with the NC worker (reasoning). The outcome can be relaxation, engagement, increase in alertness or emotional response. | *As dementia progresses, people's opportunities to engage in social interactions that are meaningful to them become more limited (context). Using their knowledge of the person's life story to develop a set of bespoke interactional tools and techniques (resources),* NC volunteers evoke an emotional response in the person (reasoning), leading to a set of relaxation, engagement and alertness outcomes. |
| 2a. | In a context where carers have seen their loved one decline and been told there is 'no hope' and received little or impersonal care (context) use of NC to evoke reactions from their loved one (resource) leads to them feeling hopeful and acknowledging that their loved one is still 'living' (reasoning). This leads to increased hope (outcome) and well-being. | Not supported. |
| 2b. | Carers provide ***task-focussed*** care and have little input from other services (context). A familiar NC volunteer provides 2 hours of interaction with the person with dementia (resource) ***which eases off worries about*** the person with dementia and allows them to have some respite (reasoning) which leads to an ***increase in well-being*** (outcome). | Carers provide ***continuous*** care and have little input from other services (context), provision of 2 hours contact with a trained NC volunteer (resource) allows them to concentrate on other things, ***knowing that the their loved one is in safe hands*** (reasoning) which gives them ***restorative time and space*** (outcome). |
| 3. | Being often task focussed, family members recognise the value of NC (context). A tailored activity box is left at the person's home (resource). The family are keen to engage in activities that enable them to connect emotionally with the person (reasoning). Family use NC independently (outcome) | Not supported |
| 4. | One volunteer is aligned to a person with dementia and spends 2 hours per week solely with that person (context). The volunteer therefore has a knowledge history of what works/doesn't work and what the person likes (resources). This allows the volunteer and the person with dementia to develop a strong emotional connection (reasoning). The outcome could be considered as the recognition of the volunteer by the person with dementia ***but actually this leads to an increased engagement which might have previously been thought of as impossible (outcome).*** | One volunteer is aligned to a person with dementia and spends 2 hours per week solely with that person (context). The volunteer therefore has a knowledge history of what works/what doesn't work and what the person likes (resources). This allows the volunteer and the person with dementia to develop a strong emotional connection (reasoning). The outcome could be considered as the recognition of the volunteer by the person with dementia ***but actually this leads to friendship, which could suggest an increased quality of life for both people (outcome).*** |

NC, The End-of-Life Namaste Care Program for People with Dementia.

## DISCUSSION

This preliminary study developed initial programme theories for the novel use of NC in peoples' own homes, as opposed to care homes. Including contrasting programme theories 2a and 2b, in total five programme theories were developed from the focus groups with NC volunteers, of these programme theories, three were supported (table 2).

The 'one-on-one' delivery of NC in the home setting in this study was highlighted by family carers as being preferable, not only because the person was in familiar surroundings but due to the increased engagement this provided. Family carers suggested that their loved one would be more likely to disengage in a group environment. NC aims to engage the senses and using it in the home setting could have the potential to allow more tailored delivery, with fewer distractions.

Evidence suggests that sustained lack of stimulation can be detrimental to people in care homes who suffer from dementia, as it augments the apathy, boredom, depression and loneliness that often accompany the progression of dementia.[38 39] The same, if not more enhanced, could be assumed for those with dementia who live at home and this could be supported by the preliminary findings of

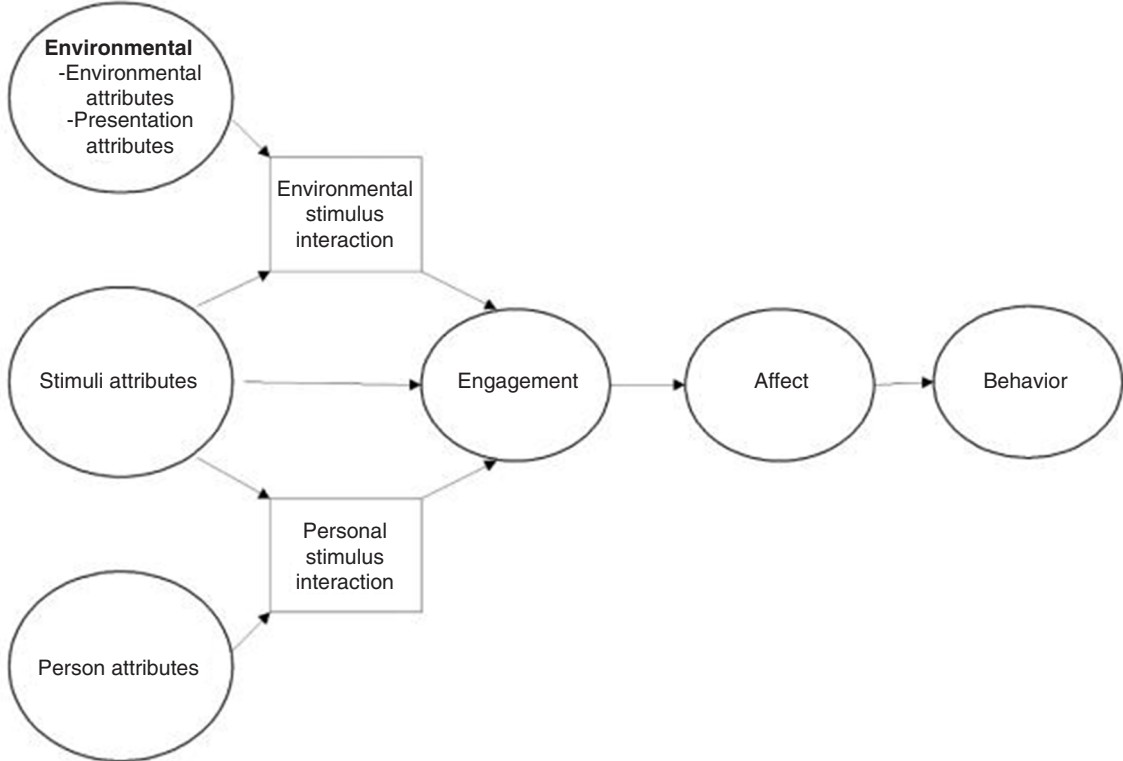

**Figure 1** Framework for engagement of people with dementia (reproduced from: Cohen-Mansfield, J., M. Dakheel-Ali, and M. Marx, engagement in persons with dementia: the concept and its measurement. American Journal of Geriatric Psychiatry 2009. 17(4): p. 299-307). Image reproduced with permission of the rights holder, Professor Cohen-Mansfield.

this research. This study and others[40 41] have highlighted the importance of social interaction for people living with dementia; those living at home with dementia have very little interaction with people other than their family and formal carers, due to issues of mobility and anxiety outside of home. Furthermore, family carers expressed an inability to interact with their loved one as they used to, this is in line with observations from another study using NC, which focussed on touch.[20] This finding could warrant further investigation in care homes also.

Cohen-Mansfield *et al*[39] suggest a framework for engagement of people with dementia (figure 1, reproduced). The theoretical framework suggests that environmental attributes (home setting), stimuli attributes (sensory activities) and person attributes (NC: Life Story, matched volunteers and continuity with volunteer), alongside interactions among these attributes, affect engagement with stimuli by the person who has dementia. NC in the home environment could be said to be more open to personalised and tailored activities than a care home environment, with a 'one-on-one' approach and less distractions, such as other residents, therefore making the environment facilitative. Stimuli presented to people with dementia in NC are also matched at first with the person's attributes, through use of the Life Story. Cohen-Mansfield *et al*[19] suggest that personalised activities are more likely to engage those with dementia.[42] This conceptual framework concerning engagement of persons with dementia therefore reflects NC well and could also be applicable

to the use of NC in care homes, as well as in people's own homes. The authors have also developed a measurement of engagement, which could potentially be used in future research on NC given their complementarity of one another.

Caring for people with dementia can be stressful, lead to family conflicts and cause burnout;[43] recent research has highlighted a need for further exploration of family carers' views about care for those with dementia at home.[44 45] One of the unintended consequences of NC in the home setting was its use by family carers for respite. NC aims to engage the family, with care home staff encouraging family and friends to join in where appropriate.[23 46] However, usual use of NC is in a care home setting, where family members do not provide the majority of task focussed care. The family carers in this study described a lack of support and a need for respite, which is supported in the literature.[47] NC provided a weekly 2 hour window of respite in which family carers could have restorative time and space. Furthermore, the findings suggested that the person with dementia may feel uncomfortable with their family member providing sensory stimulation which could be seen as placing additional time demands on family members. Future research should investigate whether those who do not know the person, such as volunteers and care home staff, are better placed to deliver NC.

This exploratory research has started to provide explanations of how NC may work in the home setting. Future

research has been briefly previously outlined, but could also include investigations into volunteer delivery of NC in care homes, to allow the intervention to also be delivered to those with milder dementia. Furthermore, an ethnographic approach to develop further understanding of outcomes for those receiving NC would be beneficial.

### Strengths and limitations

To our knowledge, this is the first formal evaluation of NC in the home setting. It is also the first to explore the use of volunteers to deliver NC. The findings highlight that further research is necessary, but outline interesting findings in terms of intervention fidelity and unintended outcomes.

A limitation of the study is the sample size; while some programme theories were not substantiated by the data, it could be that this was due to this particular sample. As in all realist research, these findings do not claim finality, but merely the beginning of an explanatory endeavour for NC.

A caution should also be outlined in interpreting the findings, due to the vast differences in implementation in people's own homes in comparison to care homes. Adapting an intervention like NC to work in the home environment does bring challenges for evaluation as the intervention itself is inevitably altered to facilitate delivery. In this delivery of the intervention, the 'dose' was different, however, recent research found little empirical evidence on the optimal 'dose' of sensory interventions. Furthermore, the interaction with volunteers as opposed to care home staff warrants further investigation and the inclusion of those with mild dementia poses questions around intervention focus and benefit, given that NC was developed for people with dementia who have physical and cognitive deterioration and are unable to engage with other activities. However, recent research highlights the challenge of examining whether the impact of interventions vary depending on cognitive ability and indicates that further research is needed to assess how psychosocial interventions can be of use across the stages of dementia.[14]

As is the process for realist research, theories were tested and refined or rejected. We aimed to report as much as possible on the process of analysis in order to be transparent and rigorous. Furthermore, it is important to counteract publication bias of only positive results, although we do not consider the unintended consequences identified in this study negative (that of respite). Finally, it also enables the research field to build on the knowledge created and discourages repeated research in the same area.

### Implications for clinicians and policymakers

The research highlights positive outcomes for people with dementia, volunteers and family members. However, it also highlights that NC may not work in the same way in the persons own home, as it does in care homes. This does not detract from the value of NC, but warrants further investigation. It also indicates the unmet needs of family carers. In order to facilitate those with dementia to live at home and to meet the current drive of care into the community, we need to first ensure the needs of those with dementia and their carers are met, whether these needs be physical, emotional or social.

### CONCLUSION

A recent cohort study indicated that people with advanced dementia still often live with distressing symptoms[48] and that community services are often not tailored to their non-medical needs.[29] Longitudinal input focussed on improving quality of life using personalised interventions such as NC shows promise in optimising life for those with dementia and also could provide much needed respite for family carers when delivered in the home setting using volunteers.

**Acknowledgements** We would like to thank all the hospice staff, volunteers and carers who took part in the research. This work was supported by Fuse (The Centre for Translational Research in Public Health), specifically the Fuse Pump Prime Fund.

**Contributors** SMD drafted the manuscript with conceptual input from ML. NK drafted sections relating to NC. JA drafted sections on palliative aspect of NC. ML, NK, JA and ST all read and commented on drafts and approved the final submitted version.

**Funding** SMD and ML are members of Fuse, the Centre for Translational Research in Public Health (www.fuse.ac.uk). Fuse is a UK Clinical Research Collaboration (UKCRC) Public Health Research Centre of Excellence. Funding for Fuse from the British Heart Foundation, Cancer Research UK, Economic and Social Research Council, Medical Research Council, the National Institute for Health Research, under the auspices of the UKCRC, is gratefully acknowledged. The views expressed in this paper do not necessarily represent those of the funders or UKCRC. The funders had no role in study design, data collection and analysis, decision to publish or preparation of the manuscript.

**Competing interests** None declared.

**Patient consent for publication** Not required.

**Ethics approval** This research was approved through Northumbria University Ethical Approval System (reference: HLSCW161705). All participants gave informed consent.

**Provenance and peer review** Not commissioned; externally peer reviewed.

**Data availability statement** No data are available.

**ORCID iD**
Sonia Michelle Dalkin http://orcid.org/0000-0002-3266-5926

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
