## [Reviewer comments · BMJ Open]

ARTICLE DETAILS

TITLE (PROVISIONAL)	Namaste Care in the home setting: Developing initial realist explanatory theories and uncovering unintended outcomes
AUTHORS	Dalkin, Sonia; Lhussier, Monique; Kendall, Nicola; Atkinson, Joanne; Tolman, Sharron

VERSION 1 – REVIEW

REVIEWER	Hanneke Smaling PhD senior researcher, Leiden University Medical Center, Leiden, Netherlands
REVIEW RETURNED	07-Aug-2019

GENERAL COMMENTS	This study contributes to the knowledge about the how, why and for whom Namaste Care has specific outcomes in the home setting. Improving quality of life and providing respite for carers in the home setting is very important. This paper therefore taps into a relevant topic! There are a few issues I would like to authors to address. Overall, they did a good job! INTRODUCTION: 1. I would like to point out to the researchers that there is already a Namaste Care service in London (UK) where they successfully have been delivering the program to people with dementia in their own home for more than 5 years (10 sessions, 1hour per session, delivered by volunteers). This could be included to the Introduction for a more complete overview. 2. It is not clear how this study will contribute to the how and why the NC outcomes in care homes occur. Table 1 provides a clear overview of the differences between NC in the care home and home setting. Their study is focused on the home setting. I therefore have doubts about whether this study will be able to provide the answers to the how and why in the care home setting. Maybe they can say something about this for the home setting. That could also be (partially) applicable for the care home setting. However, more research in the care home setting would be required. 3. While the authors indicate the novel aspects of their study, the study aims or research question(s) are missing. I would like to see them before reading about the Methods. METHODS 1. More information is needed: Did the participants receive any compensation for their participation? Could you please provide us with more information about the participants (some demographic info, like age, gender, experience of volunteer, etc.)? What were the inclusion and exclusion criteria for the participants? When could the
--

	family carers participate in the interviews (for example a minimum number of NC sessions had to be given to their relative before they could be interviewed)? 2. Not sure whether the mission and vision of the hospice are relevant for this study. It is about NC, so my suggestion would be to remove that information. 3. P11: Did any of the volunteers report “issues”? If so, how often? Could you please tell a bit more about the reported issues? 4. Please move the study objective to the end of the Introduction. FINDINGS 1. S28-37: this information is related to the implementation/delivery of the intervention. Moving it to the “setting and referrals” paragraph (p11) may be worth considering to improve the structure. 2. What does FG mean? DISCUSSION 1. I understand that carers prefer a one-on-one delivery of NC in the home setting. However, I do not see another possibility of delivering NC another way in the home setting, apart from offering it an adult day care for people with dementia. Maybe the authors can reflect a bit more on this. 2. Conclusion is generally formulated. Could you please make it more specific for NC in the home setting? Minor issues:  - Please explain the abbreviation used in the abstract (NC, p4, s21). - Please remove the first names and initials in the reference on p5 last sentence, p6 first sentence. This issue comes up multiple times, so please check the manuscript and resolve this when needed. - Please remove the abbreviation ‘NICE’ because the abbreviation is not used in the manuscript (p.6, s5). - Should it not be ‘refers’ on p6, s 25 instead of ‘refer’? - The name used for the program is not consistent (see p4, s6 and reference Simard 2013 versus p6, s41). - Same abbreviation (NC) is used twice, see p6, s41 and 51. - The abbreviation of PwD is not explained (p8). Also, PwD is considered to be not dementia-friendly. I suggest to adjust this to people living with dementia. - The “.” is missing at the end of s41
REVIEWER	Teresa Atkinson (Senior Research Fellow) Association for Dementia Studies University of Worcester Woodbury 135 Henwick Grove Worcester WR2 6AJ
REVIEW RETURNED	29-Aug-2019
GENERAL COMMENTS	This is an excellent paper and very welcome in the field of both the specialist topic (NC) and the methodology used. The findings are very clearly articulated and the four theories generated are well explained with good supporting material. I particularly like the use of Table 2 setting out the Programme

	theories. My suggestions for 'Minor Revisions' are very minor and reflect only the need for explanation to audiences not so familiar with this methodology:  1. Please explain the term retrodution on page 11 line 53 2. I think the paper would benefit from more of a link between the programme theories and the explanation of CMO's which is given on page 9. For example, Programme Theory 2a is discussed on Page 16 in terms of Context, Resource, Reasoning and Outcome. Mechanism isn't mentioned which is a little confusing for the reader. The explanation given on page 9 may need to be slightly revised to give a clearer understanding; for example C, the context, is the 'person with dementia and what they are experiencing but this isn't immediately clear within the current paragraph. Similarly, 'mechanism' may need to be mentioned on page 16. With these issues addressed I believe the paper will be an excellent contribution to both the journal and this field of research.
--	---

VERSION 1 – AUTHOR RESPONSE

Reviewer: 1	
This study contributes to the knowledge about the how, why and for whom Namaste Care has specific outcomes in the home setting. Improving quality of life and providing respite for carers in the home setting is very important. This paper therefore taps into a relevant topic! There are a few issues I would like to authors to address. Overall, they did a good job!	We would like to thank Hanneke Smaling (PhD) for their review and constructive comments.
1. I would like to point out to the researchers that there is already a Namaste Care service in London (UK) where they successfully have been delivering the program to people with dementia in their own home for more than 5 years (10 sessions, 1hour per session, delivered by volunteers). This could be included to the Introduction for a more complete overview.	We have now added this information to the introduction.
2. It is not clear how this study will contribute to the how and why the NC outcomes in care homes occur. Table 1 provides a clear overview of the differences between NC in the care home and home setting. Their study is focused on the home setting. I therefore have doubts about whether this study will be able to provide the answers to the how and why in the care home setting. Maybe they can say something about this for the home setting. That could also be (partially) applicable for the care home setting. However, more research in the care home setting would be required.	We have expanded upon how we think this research may contribute to the 'how and why' NC outcomes occur in care homes in the introduction. We have also indicated in the Discussion where we think the findings of this study need further investigation in care homes.
3. While the authors indicate the novel aspects of their study, the study aims or research question(s) are missing. I would like to see them before reading about the Methods.	The study aim has now been moved to the end of the Introduction.
1. More information is needed: Did the participants receive any compensation for their participation? Could you please provide us with more information about the participants (some demographic info, like age, gender,	Information about remuneration, volunteer experience and gender have now been provided. Data on the age of volunteers, family carers

experience of volunteer, etc.)? What were the inclusion and exclusion criteria for the participants? When could the family carers participate in the interviews (for example a minimum number of NC sessions had to be given to their relative before they could be interviewed)?	or those with dementia was not collected due to the exploratory theory generating focus of the study.
2. Not sure whether the mission and vision of the hospice are relevant for this study. It is about NC, so my suggestion would be to remove that information.	This information has now been removed.
3. P11: Did any of the volunteers report "issues"? If so, how often? Could you please tell a bit more about the reported issues?	Specific issues were not recorded per family in the study due to the focus on explaining if and how Namaste Care works, but general examples of issues have been provided. We hope this provides clarity.
4. Please move the study objective to the end of the Introduction.	This has been actioned.
S28-37: this information is related to the implementation/delivery of the intervention. Moving it to the "setting and referrals" paragraph (p11) may be worth considering to improve the structure.	This has now been moved.
What does FG mean?	Focus Group. This has now been explained in the text.
1. I understand that carers prefer a one-on-one delivery of NC in the home setting. However, I do not see another possibility of delivering NC another way in the home setting, apart from offering it an adult day care for people with dementia. Maybe the authors can reflect a bit more on this.	Further reflections on this are provided by the family carers in Phase 2, but additional information has also now been added to phase 1.
2. Conclusion is generally formulated. Could you please make it more specific for NC in the home setting?	This has now been edited to focus on the home setting and respite for carers.
Minor issues: - Please explain the abbreviation used in the abstract (NC, p4, s21). - Please remove the first names and initials in the reference on p5 last sentence, p6 first sentence. This issue comes up multiple times, so please check the manuscript and resolve this when needed. - Please remove the abbreviation 'NICE' because the abbreviation is not used in the manuscript (p.6, s5). - Should it not be 'refers' on p6, s 25 instead of 'refer'? - The name used for the program is not consistent (see p4, s6 and reference Simard 2013 versus p6, s41). - Same abbreviation (NC) is used twice, see p6, s41 and 51. - The abbreviation of PwD is not explained (p8). Also, PwD is considered to be not dementia-friendly. I suggest to adjust this to people living with dementia. - The "." is missing at the end of s41	These issues have now all been addressed. Thank you for taking the time to make note of these.
This is an excellent paper and very welcome in the field of both the specialist topic (NC) and the methodology used. The findings are very clearly articulated and the four theories generated are well explained with good supporting material. I	We would like to thank Teresa Atkinson (Senior Research Fellow) for her review, helpful comments, and for taking the time to review

particularly like the use of Table 2 setting out the Programme theories. My suggestions for 'Minor Revisions' are very minor and reflect only the need for explanation to audiences not so familiar with this methodology. With these issues addressed I believe the paper will be an excellent contribution to both the journal and this field of research.	the article.
1. Please explain the term retrodution on page 11 line 53	This term has now been explained.
2. I think the paper would benefit from more of a link between the programme theories and the explanation of CMO's which is given on page 9. For example, Programme Theory 2a is discussed on Page 16 in terms of Context, Resource, Reasoning and Outcome. Mechanism isn't mentioned which is a little confusing for the reader. The explanation given on page 9 may need to be slightly revised to give a clearer understanding; for example C, the context, is the 'person with dementia and what they are experiencing but this isn't immediately clear within the current paragraph. Similarly, 'mechanism' may need to be mentioned on page 16.	The example of CMO on page 10 in the methods section has now been updated to include resource and reasoning, to make the link between the programme theories and CMO clearer.

VERSION 2 – REVIEW

REVIEWER	Hanneke Smaling PhD Leiden University Medical Center, the Netherlands
REVIEW RETURNED	10-Oct-2019
GENERAL COMMENTS	With the made changes, this paper is an excellent contribution to the specialist topic and research field. The authors did a great job and as a reviewer I am satisfied with the adjustments they made and how the incorporated the reviewers' feedback.